# Features of Patients Receiving Extracorporeal Membrane Oxygenation Relative to Cardiogenic Shock Onset: A Single-Centre Experience

**DOI:** 10.3390/medicina57090886

**Published:** 2021-08-27

**Authors:** Dong-Geum Shin, Sang-Deock Shin, Donghoon Han, Min-Kyung Kang, Seung-Hun Lee, Jihoon Kim, Jung-Rae Cho, Kunil Kim, Seonghoon Choi, Namho Lee

**Affiliations:** 1Department of Internal Medicine, Division of Cardiology, Kangnam Sacred Heart Hospital, Hallym University College of Medicine, Beodeunaru-ro 7-gil, Yeongdeungpo-gu, Seoul 07247, Korea; blau07@hallym.or.kr (D.-G.S.); sd10060120@hallym.or.kr (S.-D.S.); homes78@hallym.or.kr (M.-K.K.); jrjoe@hallym.or.kr (J.-R.C.); choish@hallym.or.kr (S.C.); namholee@hallym.or.kr (N.L.); 2Department of Cardiothoracic Surgery, Kangnam Sacred Heart Hospital, Hallym University College of Medicine, Seoul 07247, Korea; shalang@hallym.or.kr (S.-H.L.); jkim@hallym.or.kr (J.K.); 3Department of Cardiothoracic Surgery, Hallym University Sacred Heart Hospital, Hallym University College of Medicine, Suwon 14068, Korea; kkics@hallym.or.kr

**Keywords:** cardiogenic shock, coronary artery, ECMO, mortality, myocardial infarction

## Abstract

*Background and Objectives:* Extracorporeal membrane oxygenation (ECMO) can be helpful in patients with cardiogenic shock associated with myocardial infarction, and its early use can improve the patient survival rate. In this study, we report a mortality rate-difference analysis that examined the time and location of shock occurrence. *Materials and Methods:* We enrolled patients who underwent ECMO due to cardiogenic shock related to myocardial infarction and assigned them to either a pre- or post-admission shock group. The primary outcome was the 1-month mortality rate; a subgroup analysis was conducted to assess the effect of bailout ECMO. *Results:* Of the 113 patients enrolled, 67 (38 with pre-admission shock, 29 with post-admission shock) were analysed. Asystole was more frequently detected in the pre-admission shock group than in the post-admission group. In both groups, the commonest culprit lesion location was in the left anterior descending artery. Cardiopulmonary resuscitation was performed significantly more frequently and earlier in the pre-admission group. The 1-month mortality rate was significantly lower in the pre-admission group than in the post-admission group. Male sex and ECMO duration (≥6 days) were factors significantly related to the reduced mortality rate in the pre-admission group. In the subgroup analysis, the mortality rate was lower in patients receiving bailout ECMO than in those not receiving it; the difference was not statistically significant. *Conclusions:* ECMO application resulted in lower short-term mortality rate among patients with out-of-hospital cardiogenic shock onset than with in-hospital shock onset; early cardiopulmonary resuscitation and ECMO might be helpful in select patients.

## 1. Introduction

ST-segment elevation myocardial infarction (STEMI) remains a major cause of death, worldwide, despite advances in treatment strategies [1,2]. Additionally, STEMI is associated with an increased risk of cardiogenic shock (CS), and patients presenting with both myocardial infarction (MI) and CS have a very high mortality rate [3,4]. Therefore, early revascularisation, involving techniques such as percutaneous coronary intervention or coronary artery bypass graft, is the cornerstone of the management for patients with STEMI and CS [5,6,7]. However, despite revascularisation, patients may remain hemodynamically unstable, leading to CS [7]. In such cases, mechanical circulatory support can be helpful [4,7,8]. Extracorporeal membrane oxygenation (ECMO) is a mechanical circulatory support technique that may improve the survival rate of patients requiring cardiopulmonary resuscitation (CPR) after CS [8,9,10,11]. In addition, extracorporeal CPR (ECPR), the combination of CPR with ECMO, is also performed [12]. In patients with CS accompanied by MI, studies on the effect of ECMO timing, relative to the start of hospitalisation, are lacking. Thus, we reviewed our cases of ECMO use in patients with MI accompanied by CS to determine whether there is a difference in the effectiveness of ECMO associated with the time and place of CS occurrence.

## 2. Materials and Methods

### 2.1. Study Population

We retrospectively enrolled 113 patients who underwent ECMO due to CS, between August 2013 and June 2020, at our institution. We grouped the patients according to when shock occurred: pre-admission or post-admission. The pre-admission group included patients in whom shock occurred prior to their arrival at the emergency department (ED). The post-admission group included those whose initial vital signs, upon arrival at the ED, were stable and who developed shock in the ED after their initial vital sign were checked. CS was defined as a systolic blood pressure (sBP) < 90 mmHg for >30 min or when supportive intervention was required to maintain an sBP > 90 mmHg with evidence of end-organ damage (altered mental status, urine output < 30 mL/h, or cool extremities), according to the Korea Acute Myocardial Infarction—National Institutes of Health registry [13]. Patients were excluded from the analysis if they did not undergo coronary angiography, had shock associated with a noncardiogenic cause, or were transferred to other hospitals. The study was approved by the Institutional Review Board (approval no. 2020-02-003) of Kangnam Sacred Heart Hospital (Hallym University Seoul, Korea) and was performed in accordance with the tenets of the Declaration of Helsinki.

### 2.2. Coronary Intervention, ECMO Insertion, and Control

Coronary intervention procedures and available medical therapy were performed according to standard guidelines [14,15]. During the intervention, the attending physicians determined the need for stents, thrombo-suction, and glycoprotein IIb/IIIa inhibitors. The interventional cardiologist decided on the need for ECMO, after discussion with an experienced heart team. The ECMO catheters were implanted, with angiographic guidance, by cardiovascular surgeons or interventional cardiologists. After catheter insertion, a cardiovascular surgeon controlled the ECMO; overall patient management involved the cooperation of the surgeon and cardiologists.

### 2.3. Primary Endpoint and Subgroup Analysis

The primary outcome was all-cause death within 1 month (30 days). All deaths were considered cardiac related unless an undisputed non-cardiac cause was documented. For the subgroup analysis, we grouped the participants according to whether ECMO was started as a bailout procedure, i.e., if ECMO insertion was performed prior to coronary reperfusion.

### 2.4. Statistical Analysis

Student’s *t*-test and the Wilcoxon rank test were used to evaluate the differences between the two groups. Categorical variables were analysed using the χ^2^ test and Fisher’s exact test. The clinical outcomes were calculated based on Kaplan-Meier censoring estimates, and a comparison of outcomes between the two groups was analysed using the log-rank test. Hazard ratios and 95% confidence intervals were calculated using Cox proportional regression. Statistical significance was defined as a two-sided *p*-value < 0.05. All statistical analyses were performed using R (version 4.0.5, R Foundation for Statistical Computing, Vienna, Austria).

## 3. Results

### 3.1. Baseline Characteristics

Among the 113 eligible patients, 67 were included in the final analysis, including 35 (29 males, 82.9%) in the pre-admission group and 32 (22 males, 68.8%) in the post-admission group (Figure 1). The baseline characteristics of the patients in the two groups were similar. ST-segment elevation was the most common initial electrocardiogram finding in both groups. Non-ST segment elevation, before CS, was significantly higher in the post-admission group. However, asystole and ventricular fibrillation on the electrocardiograms were more common in the pre-admission group. All cardiac markers, including creatine kinase-MB, troponin I, and B-type natriuretic peptide, were higher in the post-admission group patients, with the elevations of creatine kinase-MB being statistically significant. All patients underwent coronary angiography, and one patient in the pre-admission group underwent an emergent coronary artery bypass graft. Three patients in the pre-admission group failed reperfusion due to left ventricular rupture followed by MI, acute on chronic total occlusion of infarct-related artery (Table 1).

CPR was performed significantly more frequently on patients in the pre-admission group than in the post-admission group; the CPR duration was also longer in the pre-admission group, but the difference was not significant. The time between CS occurrence and coronary reperfusion was shorter in the post-admission group patients than in the pre-admission group; however, the difference was not statistically significant (*p* = 0.09). The time between CS occurrence and ECMO insertion was not statistically different between the two groups (*p* = 0.79). In the pre-admission group, 20 patients (57.1%) received bailout ECMO, whereas 11 (34.4%) in the post-admission group received it (*p* = 0.06). In both groups, the left anterior descending artery (LAD) was the most common culprit coronary vessel (Table 2).

### 3.2. One-Month Mortality and Related Factors

The 1-month mortality rate was lower in the pre-admission group (60.0%) than in the post-admission group (84.4%, *p* = 0.05). The cumulative mortalities were also significantly lower in the pre-admission group than in the post-admission group (Figure 2).

Cox proportional regression was performed to investigate the factors associated with differences in mortality rates between the groups; the factors investigated were age; sex; past history of hypertension, diabetes, chronic kidney disease, stroke, and/or previous percutaneous coronary intervention; history of undertaking aspirin, clopidogrel, statin, renin-angiotensin system blocker, and/or calcium channel blocker therapy; ejection fraction after revascularisation; culprit coronary lesion (left main coronary artery or LAD); CPR duration; ECMO duration; and the need for bailout ECMO. Advanced age and post-admission CS were significantly associated with increased mortality. However, male patients and ECMO durations of more than 6 days seemed protective, with significantly lower mortality rates (Table 3).

### 3.3. Subgroup Analysis of Bailout ECMO

The subgroup analysis of bailout ECMO involved the 31 patients (20 in the pre-admission group and 11 in the post-admission group) receiving this intervention. The bailout group showed better 1-month mortality results than the non-bailout group in the Kaplan–Meier analysis, although the difference was not statistically significant (Figure 3). We investigated the factors associated with bailout ECMO insertion, and found that advanced age (≥70 years), preadmission CS, absence of hypertension, and starting CRP within 9 min of the MI were beneficial; however, the benefits were not statistically significant (Appendix A). There was no between-group difference in the primary outcome according to the use of bailout ECMO (Appendix A).

## 4. Discussion

In this study, we compared the features of, and 1-month mortality rates for patients with CS subsequent to an MI, and who received ECMO before or after hospital admission. The interval between the development of CS and the application of ECMO was not significantly different between the groups. CPR was performed significantly more frequently in the pre-admission group than in the post-admission group, and the interval between CS development and CPR performance was shorter in the pre-admission group. The analysis of the patients indicated that the 1-month mortality rate was also lower in the pre-admission group. Our analysis of the factors related to the difference in mortality rates suggested that male sex and an ECMO duration of more than 6 days were statistically significant factors associated with the difference. A subgroup analysis suggested that the use of bailout ECMO did not result in a significant difference in the 1-month mortality rate, relative to the non-bailout ECMO.

In some studies, sex has not been associated with mortality rates following CS-complicated MIs, although females have been reported to have more advanced events [16,17]. In contrast, Vallabhajosyula et al. reported that female sex is an independent factor associated with poor outcomes, particularly in older patients [18]. Our report similarly showed that male sex was a factor associated with better outcomes in patients receiving ECMO due to CS. However, the existence of sex-related outcome differences remains controversial, and the reason for such differences remains unclear [16].

Out-of-hospital cardiac arrests (OHCAs) are well known to have worse outcomes than in-hospital cardiac arrests (IHCAs) [19,20,21,22,23]. However, our review of patients who received ECMO due to CS related to MIs documented better mortality outcomes for the patients in the pre-admission group than for those in the post-admission group. Several reasons may explain this difference between our study and those examining IHCAs versus OHCAs. First, the different inclusion criteria may have contributed. As mentioned above, our study included only patients with CS-complicated MIs. Second, CPR, particularly cardiac compressions, was performed more frequently in patients in the pre-admission group than in those in the post-admission group, leading to a shorter interval between CS and CPR. Finally, although not statistically significant, bailout ECMO tended to be more common in patients in the pre-admission than in those in the post-admission group. Thus, these factors might contribute to the difference in one-month mortality rates between the groups.

In our hospital setting, clinicians tend to consider the condition of patients with out-of-hospital CS as being more severe. Although CS, itself, may be a risk factor for in-hospital mortality, the patient information related to CS that might affect mortality rates, such as interval between shock occurrence and the application of CPR, may be insufficient for patients in the pre-admission group [24]. Conversely, patients in the post-admission group received full monitoring, including electrocardiograms and vital sign measurements during admission. Therefore, if shock occurred after admission, its detection and the clinical decision to administer epinephrine, CPR, or other interventions could be made earlier than for patients in the pre-admission group. Hence, these advantages for the patients in the post-admission group might interrupt the decision to initiate cardiac compressions and begin bailout ECMO. For patients in the pre-admission group, accurately determining the time of shock onset is more difficult than for patients in the post-admission group, making the use of epinephrine impossible. Hence, the decision to perform cardiac compressions may be made earlier, and clinicians might decide to begin ECMO first, rather than coronary revascularisation, in the patients in the pre-admission group. Thus, although patients in the pre-admission group might have been in CS longer than those in the post-admission group, their exposure to hypoxic injury may have been reduced; this might have resulted in the lower cumulative mortality in the pre-admission group.

ECMO is a mechanical circulatory support (MCS) technique that may help salvage patients experiencing CS or cardiac arrest [25,26]. The ECMO circuit employs a pump and an oxygenator that can provide extracorporeal gas exchange and supply oxygenated blood to the systemic circulation, promoting patient recovery [26]. ECMO has several beneficial effects in patients with MIs complicated by CS. In particular, it reduces left ventricular volumes and myocardial oxygen consumption, which may limit the infarct size [7,8]. Hence, this is part of the rationale for introducing ECPR and its reported benefits [27]. Although ECPR did not show a beneficial effect in our study, previous studies have documented that ECPR in patients with IHCAs showed better outcomes than in patients with OHCAs [9,28,29]. In addition, Wang et al. reported that ECPR might be beneficial for selected patients with OHCAs [11]. Although additional research on the effects of ECMO is necessary, research involving such MCS devices remains lacking because of the logistical and ethical limitations of research involving CS patients. Therefore, the early use of MCS devices, including intra-aortic balloon pumps, Impella^®^ heart pumps, TandemHeart^®^ ventricular assist devices, and ECMO, continues to have only a low evidence-based level recommendation in the current guidelines (Class IIb) [30,31]. Nevertheless, several studies have reported that veno-arterial ECMO might be considered for early use in selected patients with refractory CS [32,33,34,35,36]. Furthermore, a recent systematic review documented that the temporary use of veno-arterial ECMO could provide survival benefits for patients with MI-induced CS [37]. Based on our results, the application of ECMO might be helpful for some patients who develop out-of-hospital CS secondary to an MI.

Our study has several limitations. First, despite enrolling all patients who received ECMO due to post-MI CS since our centre introduced ECMO, our cohort was small. Thus, comparing the values between the groups might be inaccurate. Moreover, an initial learning period was required for physicians to become accustomed to the ECMO procedure. This may have caused the initial mortality rate to be overestimated. Second, the retrospective nature of the study necessitated dependence on the data present in the electronic medical records. Thus, some data, particularly past medical history information, might have been missing. Regardless, the significant difference in mortality rate associated with ECMO use in the pre-admission group, compared with the post-admission group, and the observation that age, male sex, ECMO duration, and in-hospital CS were statistically significant factors in both the unadjusted and adjusted models, was important. Third, similar to the above, there were some possible inaccuracies in our analysis of the electronic medical record data. When patients developed CS outside of a hospital, determining the duration of shock depended on emergency responder documents or witness statements. Conversely, when CS occurs in-hospital, wherever in the ED or during an angiogram, although there may be some delays in recording the exact time of events, the electronic medical records can be considered to be within an acceptable margin of error. Despite these limitations, our retrospective, single-centre review attempted to overcome the logistical and ethical limitations by retrospectively analysing all cases of ECMO due to CS. Furthermore, our data were selective for CS events that were accompanied with myocardial infarction, and our study reflected real-world practice, including ECPR, ECMO without CPR, and bailout ECMO. For these reasons, our study provides important clinical information about the potential benefits of ECMO for patients with MI-related CS.

## 5. Conclusions

In conclusion, the use of ECMO in patients with out-of-hospital CS might reduce mortality. In addition, early CPR in conjunction with ECMO might be helpful for select patients with CS. This study also provides the impetus for future research.

## Figures and Tables

**Figure 1 medicina-57-00886-f001:**
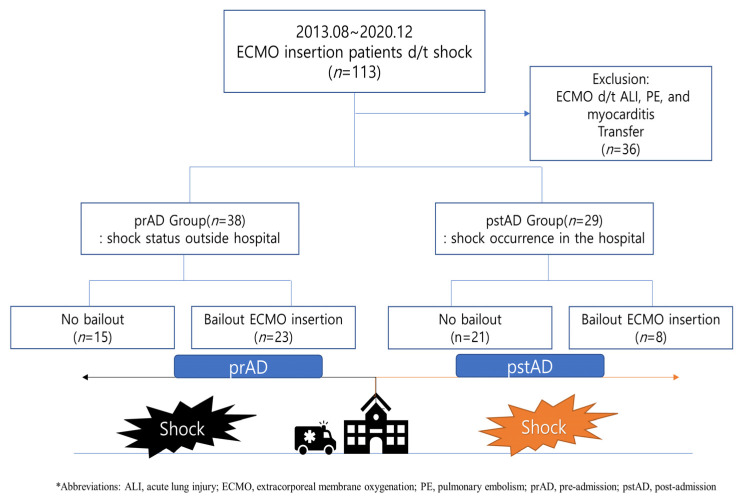
Study flow chart.

**Figure 2 medicina-57-00886-f002:**
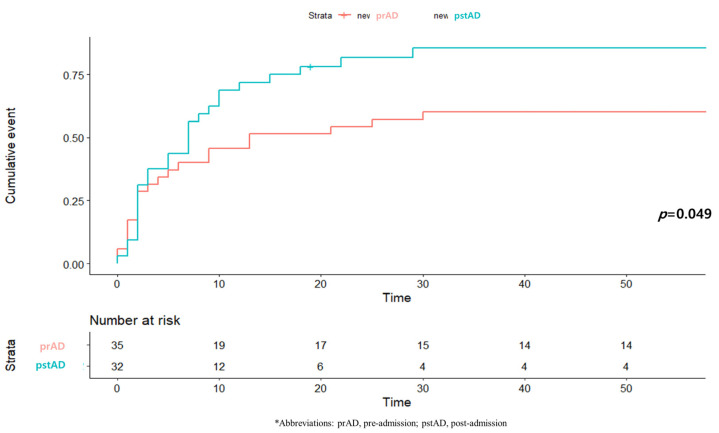
Kaplan-Meier survival plot of one-month mortality.

**Figure 3 medicina-57-00886-f003:**
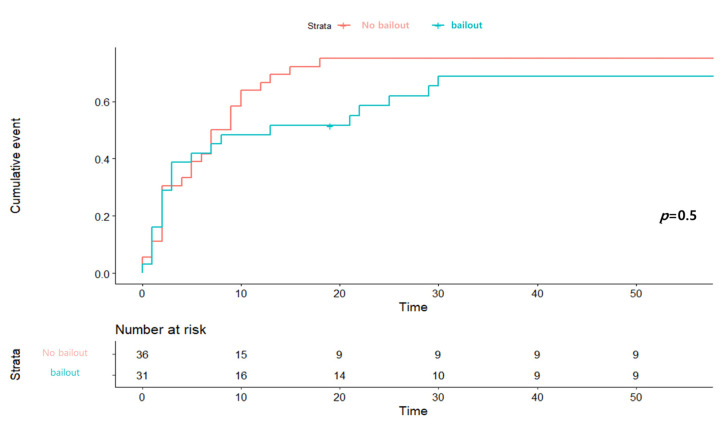
Kaplan–Meier curve of 1-month mortality in subgroup according to ECMO insertion for a bailout.

**Table 1 medicina-57-00886-t001:** Baseline patient characteristics.

	Pre-Admission Group (*n* = 35)	Post-Admission Group (*n* = 32)	*p*-Value
**Age (years)**	65.0 (55.0–73.0)	68.5 (60.5–77.0)	0.46
**Males, *n* (%)**	29 (82.9)	22 (68.8)	0.29
**BMI, kg/m^2^**	23.2 (20.7–24.7)	25.2 (22.0–27.4)	0.03
**Cardiovascular risk factors**			
**Smoking**	9 (25.7)	4 (12.5)	0.32
**Hypertension**	13 (37.1)	20 (62.5)	0.07
**Diabetes mellitus**	14 (40.0)	12 (37.5)	1.00
**Chronic kidney disease**	1 (2.9)	7 (21.9)	0.04
**Stroke**	2 (5.7)	1 (3.1)	1.00
**Previous PCI**	6 (17.1)	3 (9.4)	0.57
**Past medical history**			
**Aspirin**	11 (31.4)	4 (12.5)	0.12
**Clopidogrel**	4 (11.4)	0 (0)	0.15
**Statins**	4 (11.4)	6 (18.8)	0.62
**ACEi/ARB**	5 (14.3)	6 (18.8)	0.87
**Calcium channel blocker**	4 (11.4)	4 (12.5)	1.00
**Initial electrocardiogram**			
**ST-segment elevation**	18 (51.4)	17 (53.1)	1.00
**Non-ST-segment elevation**	2 (5.7)	14 (43.8)	0.001
**Asystole**	9 (25.7)	1 (3.1)	0.03
**Ventricular fibrillation**	4 (11.4)	0 (0.0)	0.15
**EF after revascularization**	24.0 (16.0–30.0)	19.5 (13.0–39.5)	0.64
**Laboratory findings**			
**WBC (×10^3^/uL)**	12.9 (9.8–15.0)	12.6 (8.4–16.6)	0.76
**Haemoglobin (g/dL)**	13.6 (12.8–14.9)	11.8 (10.0–13.7)	0.01
**Platelets (×10^3^/uL ^5^)**	248.0 (168.5–309.0)	196.0 (165.5–260.0)	0.10
**Creatinine (mg/dL)**	1.1 (1.0–1.4)	1.5 (1.1–2.0)	0.02
**Creatine kinase-MB (ng/mL)**	3.3 (1.8–21.7)	22.4 (3.5–97.6)	0.01
**Troponin I (pg/mL)**	8.2 (1.1–162.1)	71.8 (2.0–5981.4)	0.06
**BNP (pg/dL)**	161.2 (39.9–292.4)	326.6 (55.1–855.5)	0.58
**Inotropes**			
**Dopamine**	30 (85.7)	29 (90.6)	0.81
**Dobutamine**	7 (20.0)	9 (28.2)	0.57
**Norepinephrine**	31 (88.6)	25 (78.1)	0.41
**Coronary reperfusion strategy**			
**PCI**	31 (88.6)	32 (100.0)	0.15
**CABG**	1 (2.8)	0 (0.0)	1.00
**Fail or none**	3 (8.6)	0 (0.0)	0.27

EF, ejection fraction; PCI, percutaneous coronary intervention; WBC, white blood cell count. Values are presented as median [interquartile range] or *n* (%). Abbreviations: ACEi, angiotensin-converting enzyme inhibitor; ARB, angiotensin receptor blocker; BMI, body mass index; BNP, B-type natriuretic peptide; CABG, coronary artery bypass graft.

**Table 2 medicina-57-00886-t002:** Shock related factors in the pre- and post-admission groups.

	Total (*n* = 67)	Pre-Admission Group (*n* = 35)	Post-Admission Group (*n* = 32)	*p*-Value
**CPR**	44 (65.7)	28 (80.0)	17 (53.1)	0.02
**CPR duration (min)**	32.5 (14.5–53.5)	43.5 (18.0–55.5)	30.0 (13.0–43.0)	0.19
**Shock to CPR interval (min)**	8.5 (1.5–55.0)	5.0 (0–14.5)	50.0 (7.0–237.0)	<0.01
**Shock to ECMO interval (min)**	92.0 (54.0–148.0)	92.0 (63.5–148.0)	90.0 (51.0–190.0)	0.79
**Shock to reperfusion interval (min)**	65.0 (1.5–113.0)	69.0 (47.0–106.5)	35.0 (−139.0–115.5)	0.09
**ECMO duration**	5.0 (2.0–8.0)	5.0 (2.0–7.0)	6.0 (2.0–8.5)	0.37
**Bailout ECMO**	31 (46.3)	20 (57.1)	11 (34.4)	0.06
**Culprit site**				
**LM**	9 (13.4)	6 (17.1)	3 (9.4)	0.57
**LAD**	34 (50.7)	17 (48.6)	17 (53.1)	0.90
**LCx**	6 (9.0)	3 (8.6)	3 (9.4)	1.00
**RCA**	18 (26.9)	9 (25.7)	9 (28.1)	1.00
**Hospital duration**	10.0 (3.0–23.5)	11.0 (2.0–25.5)	9.0 (5.0–21.0)	0.95
**1-Month mortality**	48 (71.6)	21 (60.0)	27 (84.4)	0.03

Values are presented as median [interquartile range) or *n* (%). Abbreviations: CPR, cardiopulmonary resuscitation; ECMO, extracorporeal membrane oxygenation; LAD, left anterior descending artery; LCx, left circumflex artery; LM, left main; min, minutes; RCA, right coronary artery.

**Table 3 medicina-57-00886-t003:** Hazard ratios for the factors possibly related to the mortality rate.

Values	Hazard Ratio			
	Univariable Analysis	*p*-Value	Multivariable Analysis	*p*-Value
**Age (≥70 years)**	2.22 (1.25–3.94)	0.01	2.07 (1.15–3.73)	0.02
**Male sex**	0.34 (0.18–0.63)	<0.01	0.38 (0.19–0.70)	<0.01
**Hypertension**	0.99 (0.56–1.74)	0.97		
**Diabetes mellitus**	0.98 (0.55–1.76)	0.95		
**Chronic kidney disease**	1.00 (0.43–2.36)	0.99		
**Stroke**	0.37 (0.05–2.68)	0.32		
**Previous PCI**	0.94 (0.42–2.09)	0.88		
**Aspirin**	0.59 (0.29–1.23)	0.16		
**Clopidogrel**	1.02 (0.32–3.33)	0.96		
**Statins**	0.95 (0.43–2.12)	0.90		
**ACEi/ARB**	0.61 (0.26–1.44)	0.26		
**CCB**	0.37 (0.11–1.19)	0.09		
**Post-revascularization EF (** **≥** **30%)**	0.79 (0.40–1.60)	0.52		
**CPR start (≤9 min)**	0.89 (0.47–1.73)	0.73		
**CPR duration**	1.01 (0.99–1.02)	0.14		
**ECMO duration (≥6 days)**	0.43 (0.24–0.77)	0.005	0.38 (0.21–0.72)	0.01
**PCI to LM or LAD**	1.15 (0.63–2.09)	0.66		
**Bailout ECMO**	0.82 (0.46–1.45)	0.50		
**In-hospital shock**	1.77 (0.99–3.16)	0.053	2.02 (1.07–3.81)	0.03

ACEi, angiotensin-converting enzyme inhibitor; ARB, angiotensin receptor blocker; CCB, calcium channel blocker; CPR, cardiopulmonary resuscitation; EF, ejection fraction; ECMO, extracorporeal membrane oxygenation; LAD, left anterior descending artery; LM, left main; min, minutes; PCI, percutaneous coronary intervention; RCA, right coronary artery.

## Data Availability

The datasets used and/or analysed during the current study are available from the corresponding author on reasonable request.

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
