# Peer review of "Features of Patients Receiving Extracorporeal Membrane Oxygenation Relative to Cardiogenic Shock Onset: A Single-Centre Experience"

_medicina, 2021, doi:10.3390/medicina57090886_

Round 1

Reviewer 1 Report

Comments to the Authors:

Dear Sir/Madam,

The manuscript, entitled” „Features of patients receiving extracorporeal membrane oxygenation relative to cardiogenic shock onset: A single-centre experience” is significant and noteworthy, but it requires substantial revision. It is worth pointing out that the work presented for review presents very promising results on patients with cardiogenic shock associated with myocardial infarction. Early application of ECMO can improve patient survival.

There are a number of problems with this paper:

Major comments:

  1. In the Statistical analysis section, there is no information about checking the normality of the data distribution. If the results of the discussed parameters do not show the characteristics of normal distribution, then the mean value and standard deviation should not be used. In this case, the median and the interquartile range should be used.

Substantive corrections:

  1. Parameter for evaluating kidney function is creatinine and not creatine. This wrong name was used in Table 1. This should be corrected.
  2. In their work, the Authors examined cardiological markers, including: creatine kinase MB (5 Results - Baseline characteristics). They did not specify whether they tested the activity of this enzyme or the concentration of CK-MB. However, they gave in Table 1, i.e. the unit of this parameter: ng/mL. Hence, they should introduce the addition: CK-MB mass.
  3. At this point, I wish to emphasize the importance of my major commentary, noting the statistical analysis. Please note the very large scatter in the results of cardiac markers (CK-MB, Troponin I, BNP). The median and the interquartile range should be used, not the mean value and standard deviation. This needs to be changed.

Minor comments:

  1. The Abstracts’ headings should be prepared according to the Instructions for Authors.
  2. Manuscript sections should be titled according to the Instructions for Authors.
  3. Manuscript sections numbering should be corrected according to the Instructions for Authors.
  4. Abbreviations should be included in the Figure 1 legend.
  5. In the Table 2 the values of interquartile range should be placed in square brackets.
  6. References should be prepared according to the Instructions for Authors.
  7. In the main text, reference numbers should be placed in square brackets and positioned before the punctuation.
  8. Empty cells in first column in the Supplementary Table S1 should be filled.
  9. Repeating word “mortality” in the Supplementary Figure S1 A should be removed.

Yours faithfully,

Author Response

Thank you for the valuable comments. I answered  in the word file. 

Reviewer 2 Report

i suggest to authors as major concern the need to better describe clinical characteristic of enrolled patients according to aclinical score well documented for the critical care medicine as APACHE II. 

Author Response

(The authors gave the same response as above.)
